# Tailoring the resolution of single-cell RNA sequencing for primary cytotoxic T cells

Kristiyan Kanev [1,4✉], Patrick Roelli[1,2,4], Ming Wu[1], Christine Wurmser[3], Mauro Delorenzi [2], Michael W. Pfaffl[1] & Dietmar Zehn [1✉]

Single-cell RNA sequencing in principle offers unique opportunities to improve the efficacy of contemporary T-cell based immunotherapy against cancer. The use of high-quality single-cell data will aid our incomplete understanding of molecular programs determining the differentiation and functional heterogeneity of cytotoxic T lymphocytes (CTLs), allowing for optimal therapeutic design. So far, a major obstacle to high depth single-cell analysis of CTLs is the minute amount of RNA available, leading to low capturing efficacy. Here, to overcome this, we tailor a droplet-based approach for high-throughput analysis (tDrop-seq) and a plate-based method for high-performance in-depth CTL analysis (tSCRB-seq). The latter gives, on average, a 15-fold higher number of captured transcripts per gene compared to droplet-based technologies. The improved dynamic range of gene detection gives tSCRB-seq an edge in resolution sensitive downstream applications such as graded high confidence gene expression measurements and cluster characterization. We demonstrate the power of tSCRB-seq by revealing the subpopulation-specific expression of co-inhibitory and co-stimulatory receptor targets of key importance for immunotherapy.

[1] Division of Animal Physiology and Immunology, School of Life Sciences Weihenstephan, Technical University of Munich, 85354 Freising, Germany. [2] BCF, Swiss Institute of Bioinformatics, University of Lausanne, 1015 Lausanne, Switzerland. [3] Division of Animal Breeding, School of Life Sciences Weihenstephan, Technical University of Munich, 85354 Freising, Germany. [4] These authors contributed equally: Kristiyan Kanev, Patrick Roelli. ✉email: kanev@wzw.tum.de; dietmar.zehn@tum.de

Single-cell RNA sequencing (scRNA-seq) developed into the method of choice to obtain an unbiased high-resolution snapshot of the ad hoc gene expression programs used in individual cells. Compared to bulk population sequencing, there are several key advantages provided by single-cell resolved gene expression profiles (scGEPs). These include the ability to deconvolute cellular heterogeneity in mixed populations, to extract gene expression networks, and to identify regulatory relationships between genes based on truly occurring co-expression within the same cell. Moreover, the scGEPs provide the unique opportunity to track trajectories of differentiation and progenitor–progeny relationships between cells[1]. Altogether, this is crucial for improving our understanding of the development and differentiation of T cell populations with diverse function and phenotype. A deeper knowledge in mechanisms orchestrating this complex differentiation process is urgently needed to direct the next generation of immunotherapeutic approaches.

In an immune response, single naive T lymphocytes bearing unique antigen receptors recognize their cognate antigen and activate, rapidly giving rise to 1000–10,000 clonally expanded daughter cells[2,3]. The resulting cellular progeny bear the same antigen-specific receptor but develops into heterogeneous subpopulations with specialized developmental and functional potential[4]. So far, it is well established that activated T cell populations contain at least two distinct subsets—terminally differentiated effector cells, which control the ongoing infection, and progenitor cells which retain proliferative capacity and plasticity. The latter express the transcription factor Tcf1, which can be used for their identification[5]. In infections resolved by the immune system, the progenitors differentiate into memory T cells. If the antigen persists (e.g., chronic infection), the progenitors serve a reservoir function by constantly supplying newly generated short-lived effector cells[6–11]. Thus the progenitors are considered a key population for therapeutic interventions, since effective targeting and activation of this subset appears to be a pre-requisite to install protective or curative immunity in chronic infection or cancer. In addition, several effector T cell populations with varying functional potential have already been identified in chronic infections[12–15]. Thus it is of utmost importance for immunotherapies to identify potent targets and strategies that selectively manipulate the dynamics of specific T cell subpopulations. Single-cell gene expression profiling offers unique opportunities in this respect. Despite the extensive use of scRNA-seq in the field of immunology, a key limitation is that the typical protocols struggle with the particularities of T cells, which, in contrast to other cells, contain only minute amount of RNA. Thus there is an urgent need to develop T cell-tailored solutions with improved mRNA capturing efficacy.

In this work, we use a well-established relevant experimental system to obtain naive or differentiated T cell populations and perform a series of optimizations of the classical droplet sequencing (Drop-seq)[16] and single-cell RNA barcoding and sequencing (SCRB-seq)[17]. Thus we establish T cell-tailored variants of both protocols designated as tDrop-seq and tSCBR-seq (t from T cell). tDrop-seq is a tool for cost-effective high-throughput but shallow single-cell transcriptome profiling of cytotoxic T cells, which is highly valuable for initial exploratory analysis. tSCBR-seq is a tool with superior power to delineate fine transcriptomic differences between transcriptionally similar cytotoxic T lymphocyte (CTL) populations. The power of the latter one is a result of its superior mRNA capturing efficacy when compared to commonly used droplet-based methods (detects 17- and 12-fold more transcripts per gene than tDrop-seq and 10xGenomics Chromium, respectively). Finally, using tSCBR-seq we identify compartment-specific

regulatory receptors, which could be used for selective therapeutic targeting of progenitor, functional, and dysfunctional cytotoxic T cells.

## Results

**Experimental systems and set-up.** Drop-seq and SCBR-seq are currently two of the most prominent methods for scRNA-seq. The former is cost efficient and high throughput. The latter has a high power to detect differentially expressed genes, due to the high number of captured transcripts per gene[18]. Both methods incorporate unique molecular identifiers (UMIs), which allows for absolute quantification of gene expression by effectively eliminating the bias introduced by PCR (Supplementary Fig. 1C)[19–21]. In order to optimize both protocols for primary T cells, we utilized a highly standardized experimental system to obtain naive or in vivo differentiated T cell populations. This system relies on P14 T cell receptor (TCR) transgenic CD8 T cells, which recognize the gp33 epitope of the commonly used in mouse infection models lymphocytic choriomeningitis virus (LCMV; Supplementary Fig. 1A). In a typical experiment, naive P14 T cells are obtained from transgenic donor mice and transferred in low numbers into recipient mice. The P14 cells carry a congenic marker that is recognized by specific antibodies. This allows for convenient identification and isolation of the transferred cells in the host mice. The recipient mice are subsequently infected with a strain of LCMV causing either acute (strain Armstrong) or chronic infection (strain clone 13), which induces P14 activation and acute or chronic infection-specific differentiation programs. Prior to infection, naive (unstimulated) P14 T cells represent a biologically and transcriptionally homogeneous population, which due to cell size and content uniformity is useful to assess technical performance between the two protocols and their optimizations. In an acute infection with LCMV Armstrong, the P14 T cells develop a fully functional effector phenotype that is able to clear the infection. This state is particularly useful to assess the detection efficacy for key immune genes necessary for CTL function among the protocols. Following a chronic infection with LCMV clone 13, the P14 T cells develop a phenotype with reduced functionality and limited ability to control the viral infection, a phenomenon known as T cell exhaustion. This state is highly informative for understanding the mechanisms suppressing the effector function of T cells, thus it can be used to interrogate the subpopulation-specific expression of relevant receptors with immunotherapeutic potential.

**Microdroplet-based techniques have inherently low mRNA capturing efficiency for primary CD8 T cells.** Due to the high number of cells necessary for profiling, the identification of rare cells within mixed populations requires a cost-effective and high-throughput scRNA-seq protocol. These requirements are met by Drop-seq[16] and its commercial analog the Chromium system from 10xGenomics, both of which are microdroplet techniques for scRNA-seq that rely on encapsulating single cells with uniquely barcoded beads into tiny droplets (Supplementary Fig. 1B)[16]. The droplets represent aqueous compartments formed by precisely combining aqueous and oil flows into a microfluidic device with the ultimate goal of capturing an individual cell and a single barcoded bead into one droplet to retain single-cell resolution. Due to its open source nature, we decided to assess the suitability of Drop-seq for high-breath CTL analysis. Initially, we performed the typical control mixing experiment of mouse and human cells, but instead of cultured cells we used primary human and mouse lymphocytes (Fig. 1A–C). The data show the successful separation of mouse from human lymphocytes as the

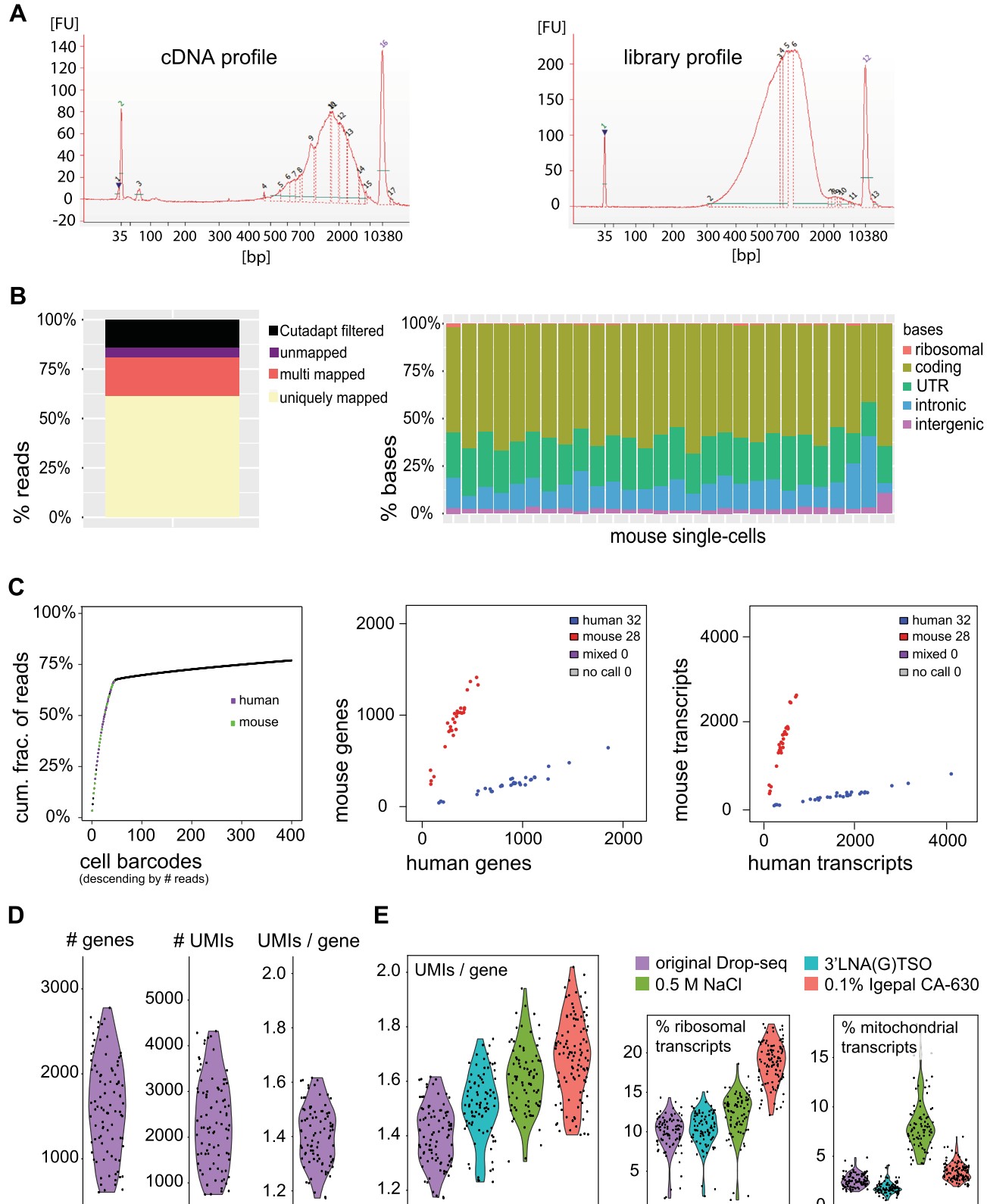

**Fig. 1 Tailoring the chemistry of Drop-seq increases its sensitivity for primary CTLs. A–C** Analysis of Drop-seq generated single-cell transcriptomes from human and mouse lymphocytes. **A** Bioanalyzer electropherograms of the generated cDNA (left) and library (right). **B** Read and base mapping statistics. **C** The knee plot represents the cumulative fraction of reads attributed to real cell and empty barcodes. The dot plots depict cells identified as singlets (aligned either to human or mouse) and doublets (having mixed human–mouse expression profile). **D**, **E** Analysis of Drop-seq generated single-cell transcriptomes from naive P14 T cells. **D** Sensitivity of the original Drop-seq protocol. **E** Sensitivity of introduced single modifications in the original Drop-seq protocol.

doublet rate, which indicates droplets that contained both mouse and human cell, was kept at zero (Fig. 1C). Thus, in this set-up, the single-cell resolution of Drop-seq was comparable to protocols based on sorting single cell into individual wells of a PCR plate (e.g., SCRB-seq), where the cell doublets are eliminated by the gating strategy. Next, we focused on assessing the sensitivity of the originally published Drop-seq protocol. Therefore, we generated single-cell gene expression profiles from naive P14 CD8 T cells. We were able to detect a median of 1607 genes and 2235 transcripts (UMIs) per cell, where a gene was detected on average with 1.4 captured transcripts (Fig. 1D). We anticipated a lower sensitivity of the unmodified Drop-seq protocol for T cell analysis compared to plate-based alternatives, but we were rather surprised to see the magnitude by which the low RNA content of CTLs negatively impacted the yield of the unmodified Drop-seq protocol.

**Tailoring the chemistry of Drop-seq moderately increases its sensitivity for primary CTLs**. Our data suggest that microdroplet-based techniques such as Drop-seq have inherently low mRNA capturing efficiency for primary cytotoxic T cells, therefore we sought to modify the chemistry to increase yield and improve performance. To achieve this, we modified the lysis and the hybridization conditions. Additionally, we tested three different reverse transcriptases (RTs) and PCR amplification in the presence of 4% Ficoll PM-400 as a macromolecular crowding agent. As indicated by the UMIs/gene ratio (Fig. 1E), we were able to moderately improve the sensitivity of the Drop-seq protocol for primary CTLs by: (1) replacing the originally used for lysis Sarkosyl detergent with 0.1% Igepal CA-630; (2) supplementing the lysis buffer with 0.5 M NaCl for increased hybridization; (3) replacing the 3′ most rG in the template switching oligo (TSO) with a locked nucleic acid base (3′LNA) to stabilize the TSO-mRNA dimer. We observed that a gene was detected on average with 1.5 (use of Igepal CA-630), 1.6 (NaCl supplementation), and 1.7 (use of 3′LNA TSO) UMIs, which was also accompanied by increased cDNA yields following PCR amplification (Supplementary Fig. 2A). From the three RTs tested, Maxima H Minus RT (ThermoFisher) and SuperScript IV RT (ThermoFisher) performed similarly well in terms of cDNA yield following PCA amplification (Supplementary Fig. 2B), so we decided to keep the originally used Maxima H Minus RT. We also observed that supplementing the PCR amplification reaction with the molecular crowding agent Ficoll PM-400 increased the cDNA yield (Supplementary Fig. 2C). Overall, we devised a T cell-adjusted Drop-seq protocol (tDrop-seq) that has increased sensitivity for primary CTLs.

**The CTL-optimized tSCRB-seq has superior mRNA capturing efficacy**. While cost efficacy and high-throughput capacity are the major benefits of the tDrop-seq protocol, the low copy number by which individual genes are detected with this method significantly limits the power of the bioinformatic analysis that can be performed with such data. In fact, higher-resolution data which delineate fine dynamic differences of gene expression are essential for several types of bioinformatics approaches, such as molecular network generation, developmental trajectory analysis, and the fine distinction between closely related subsets. As these types of analysis are critical for defining the mechanisms of T cell differentiation, having an approach with high mRNA capturing efficacy at hand will allow studying the transcriptional particularities between progenitors formed in acute and chronic infection, as well as the different effector cell subpopulations formed in functional and exhausted

T cell responses. We therefore decided to assess the suitability of SCRB-seq[17] for sensitive CTL analysis. SCRB-seq is a plate-based protocol for single-cell RNA-sequencing, which relies on sorting single-cell using fluorescent-activated cell sorting (FACS) into individual wells of a PCR plate (Supplementary Fig. 1B). In order to evaluate the sensitivity of the original SCRB-seq protocol, we attempted to generate cDNA from naive P14 CD8 T cells, but we failed to detect successful amplification with Bioanalyzer (Fig. 2A). We attributed this to the use of silica-based spin columns in the original protocol for post reverse transcription pooling of the already barcoded single-cell transcriptomes. In our experience, the spin columns for isolation of RNA and DNA have lower recovery rate, higher contamination rate, and give RNA with lower RNA integrity number than magnetic bead-based purification. To overcome this issue, we introduced a step of RNA purification before reverse transcription with the use of Agencourt RNAClean XP magnetic beads (Beckman Coulter). This step not only ensured optimal conditions for reverse transcription but also excluded potential genomic contamination. To prevent loss of valuable transcripts, we opted out of pooling the already barcoded single-cell reactions before cDNA amplification, which would have required an additional step of bead-based purification to reduce the reaction volume. Thus we performed cell-separated cDNA amplification. After amplification, cDNA was pooled and purified with the use of Agencourt AMPure XP magnetic beads (Beckman Coulter). The above described strategy yielded high-quality amplified cDNA form primary cytotoxic T cells (Fig. 2B). After library preparation and sequencing, this modification of SCRB-seq detected nearly 18-fold higher number of UMIs per gene than the original Drop-seq protocol (Figs. 2C and 1D), underlining its superior sensitivity. Since RNA purification before reverse transcription allowed for the use of harsher lysis conditions, we replaced the originally used Phusion HF buffer (1:500 dilution) supplemented with Proteinase K with a more stringent lysis solution containing 0.2% Triton X-100 detergent or TCL buffer (Qiagen) supplemented with 1% β-mercaptoethanol. We observed that, from the three lysis conditions tested, the use Qiagen TCL buffer supplemented with 1% β-mercaptoethanol improved the mRNA capturing efficacy most significantly (Fig. 2D). The median number of detected genes increased from 1350 to 1641 and the number of transcripts from 34,657 to 87,037. This was accompanied with an increase of the median number of UMIs detected per gene from 26 to 53 (Fig. 2E). Furthermore, combining the TCL based lysis with 3′LNA TSO additionally increased the median number of detected genes, transcripts (UMIs), and transcripts per gene (UMIs/gene) to 1936, 127,963, and 65, respectively. We adopted this CTL-optimized version of the SCBR-seq protocol to which we refer as tSCRB-seq (from T cells). tSCRB-seq is characterized with significantly high mRNA capturing efficacy, which allows for detection of a broader dynamic range of gene expression in CTLs.

**tSCRB-seq is characterized by higher mRNA yield and lower portion of non-informative ribosomal transcripts**. As a next step, we directly tested the ability of tDrop-seq, tSCRB-seq, and 10xGenomics Chromium to decipher immune responses side by side. For this purpose, we generated single-cell gene expression profiles from P14 cells recovered on day 8 post an acute LCMV Armstrong with tDrop-seq and tSCRB-seq, which were compared to a published 10xChromium dataset with matching experimental set-up[22]. At this time point, the recovered cells have pronounced effector phenotype, which is characterized by the expression of a well-defined set of effector molecules of key

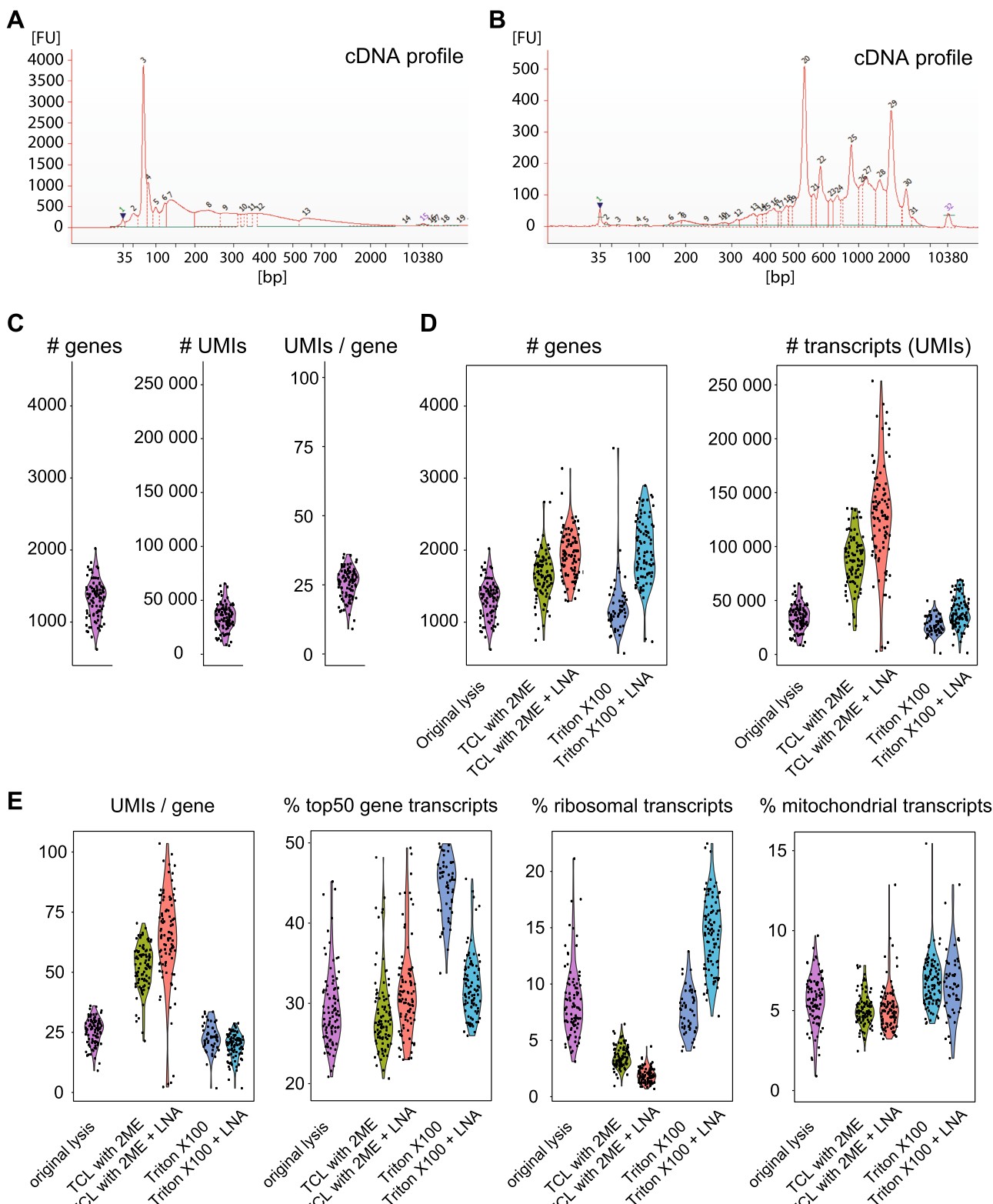

**Fig. 2 The CTL optimized tSCRB-seq has superior mRNA capturing efficacy.** Analysis of SCRB-seq generated single-cell transcriptomes from naive P14 T cells. **A** Bioanalyzer electropherogram of the cDNA profile of the original SCRB-seq protocol. **B** Bioanalyzer electropherogram of the cDNA profile of an optimized version of SCRB-seq using RNA purification before reverse transcription. **C–E** Violin plots depicting key performance parameters of different SCRB-seq modifications. Each dot represents a single cell. **C** Sensitivity of the SCRB-seq protocol with introduced RNA purification before reverse transcription. **D** Sensitivity of additional modifications of the SCRB-seq protocol with introduced RNA purification before reverse transcription. **E** Comparison of the key technical parameters among the different modifications of tSCBR-seq.

**Table 1 Compared to microdroplet techniques, tSCRB-seq is characterized by higher transcript yield and lower portion of non-informative ribosomal transcripts.**

**P14s recovered on day 8 post-acute infection**

|  | tDROP-seq | tSCRB-seq | 10× Chromium |
|---|---|---|---|
| Md. # transcripts | 4033 | 46,228 | 6708 |
| Md. # genes | 1685 | 1110 | 2030 |
| Md. % ribo. | 13% | 2% | 26% |
| Md. # ribo. | 493 | 914 | 1779 |
| Md. % mito. | 10% | 5% | 2% |
| Md. # mito. | 386 | 2235 | 159 |
| Md. % top50 | 28% | 39% | 31% |
| Md. transcripts/gene | 2.4 | 39.9 | 3.3 |

Analysis of libraries generated with tDrop-seq and tSCRB-seq from P14 T-cells recovered on day 8 post-acute LCMV Armstrong infection, compared to a published 10× Chromium dataset with matching experimental setup[22]. The table depicts key technical parameters of the libraries generated with the three methods.
*md. # transcripts* median number of captured transcripts per cell, *md. # genes* median number of captured genes per cell, *md. % ribo.* median portion of ribosomal transcripts out of the total transcripts per cell, *md. # ribo.* median number of ribosomal transcripts per cell, *md. % mito.* median portion of mitochondrial transcripts out of the total transcripts per cell, *md. # mito.* median number of mitochondrial transcripts per cell, *md. top50* median portion of transcripts attributed to the top 50 most highly expressed genes per cell, *md. transcripts/gene* median number of transcripts detected per gene.

importance for cytotoxic T cell function[4]. The 10xChromium displayed increased mRNA capturing efficacy compared to tDrop-seq (Table 1). However, tSCRB-seq provided superior mRNA capturing efficacy by detecting 17- and 12-fold more transcripts per gene than tDrop-seq and 10xChromium, respectively. Interestingly, both 10xChromium and tDrop-seq were characterized by high portions of non-informative ribosomal transcripts resulting in waste of sequencing reads (Table 1 and Supplementary Fig. 3). Compared to the transcript-rich tSCRB-seq libraries, the transcript-poor libraries of tDrop-seq and 10xChromium ensured detection of a high number of genes per cell base. Nevertheless, this did not affect the detection of the key for this time point immune genes, which were detected in similar fraction of cells generated among all methods (Supplementary Table 1). Next, we looked at how the mean number of detected transcripts per cells is affected by the sequencing depth (Fig. 3A). Both tDrop-seq and 10xChromium saturated early at comparatively low sequencing depth (about 40,000 mapped reads per cell), while tSCRB-seq reached transcript saturation at higher sequencing depth (about 120,000 mapped reads per cell). This observation matches the 10xGenomics' recommended sequencing depth of about 50,000 reads per cell for peripheral blood mononuclear cells (part of which are CD8 T cells). We recommend sequencing the tSCBR-seq-generated transcriptomes at sequencing depth of at least 200,000 reads per cell. Interestingly, tSCBR-seq captured more transcript per cell than tDrop-seq and 10xChromium even at the same sequencing depth. Next, we wanted the assess whether the observed gain of transcripts with tSCBR-seq is relevant for the detection of immune signatures (Fig. 3B). Compared to tDrop-seq and 10xChromium, tSCBR-seq captured significantly higher number of transcripts per cell of key immune genes, including transcriptional and epigenetic regulators. Moreover, tSCBR-seq detected those genes with a higher standard deviation among cells even if down-sampled to 40,000 reads per cell, which indicates a higher dynamic range of gene expression (Supplementary Table 2). Taken together, we foresee that the superior dynamic range of transcripts detected per gene with tSCRB-seq would have a critical impact on all downstream applications requiring high precision.

**tSCRB-seq enables compartment-resolved expression of key co-inhibitory and co-stimulatory receptor targets.** It is well established that the stem-like progenitor population is crucial for T cell expansion after inhibitory receptor blockade[7,23], but the regulatory receptors expressed by this population remain vaguely defined. Moreover, recent studies recognized that a highly effective immunotherapy would require more than a simple expansion of effector cells, which later acquire a debilitating exhausted phenotype (as in the case of programmed cell death protein 1 (PD-1) blockade alone), but an approach that ensures the generation and maintenance of a functional progeny[24,25]. This can be achieved by combining PD-1 blockade with a secondary treatment, aimed at promoting either progenitor or effector T cell health. Thus identifying compartment-specific expression of co-inhibitory and co-stimulatory receptors on CTLs would strongly benefit the growing field of immunotherapy, which has evolved into a serious treatment option for the millions of people suffering from malignant diseases and chronic viral infections worldwide. To provide a map of such CTL compartment-specific expression of co-inhibitory and co-stimulatory receptors for feature therapeutic strategies, we utilized a tSCRB-seq-generated dataset of about 1700 P14 T cell transcriptomes recovered at day 40 post chronic LCMV clone 13 infection from control (860 cells) and CD4-depleted animals (860 cells)[12]. In order to perform unbiased grouping of cells into clusters based on transcriptome similarities, we first used non-linear dimensionality reduction (*t*-distributed Stochastic Neighbor Embedding (tSNE)), which aims to place cells with similar local neighborhoods in high-dimensional space together in low-dimensional space. Then we used Seurat to construct graph-based clusters—cell color, which colocalized with tSNE clusters—cell location (Fig. 4). We identified five clusters, one represents the stem-like progenitors (expression of Tcf7) and four effector clusters (expression of Gzma, Gzmb, Gzmk, and Fasl), of which one with functional (expression of Tbx21 and Cx3cr1) and three with varying degree of dysfunctional phenotype (Nr4a2, Pdcd1, and CD160)[7,26,27]. Moreover, we were able to identify compartment-specific regulatory receptors, which could be used for selective targeting of progenitor (Cd9, Icos, and Tnfrsf4), functional (Il18r, Klrc1, Klrd1, and Klrk1), and dysfunctional (Cd244 and Tnfrsf9) CD8 T cells. When compared to P14 T cell transcriptomes recovered from a similar experimental set-up and generated with 10xChromium[13], the tSCRB-seq-generated transcriptomes provided more detailed and graded differential expression of key immune genes among the clusters (Supplementary Fig. 4). This particularly affected key transcriptional factors (e.g., Tbx21 and Irf7) and exhaustion markers (Pdcd1, Cd160, and Entpd1). This demonstrates the power of tSCRB-seq as highly efficient mRNA capturing protocol to delineate the fine gene expression differences among therapeutically critical CTL subpopulations.

In conclusion, we highlight the importance of optimization and consideration of the method used as a prerequisite for the successful application of scRNA-seq strategy to properly resolve the intricate relationship of cytotoxic T cell subsets in health and disease. A key decision that must be made upfront is whether high throughput and low cost or high resolution are the priorities of the analysis. In this work, we provide tools to address both needs. The cost-effective tDrop-seq is a droplet-based protocol, which can be applied in settings requiring cost-efficient analysis of high number of T cells, such as identification of rare cell populations. The major downside of tDrop-seq is the low copy number by which individual genes are detected, limiting its use in settings requiring high-power bioinformatics analysis. In contrast to the microdroplet techniques, tSCRB-seq is a plate-based protocol with superior mRNA capturing efficacy. This allows the detection of a broader dynamic range of gene expression, which

**A**

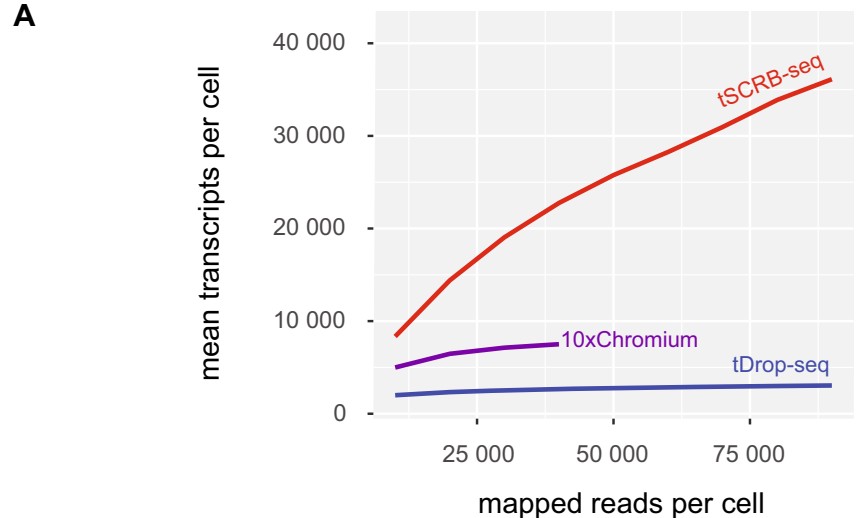

**B**

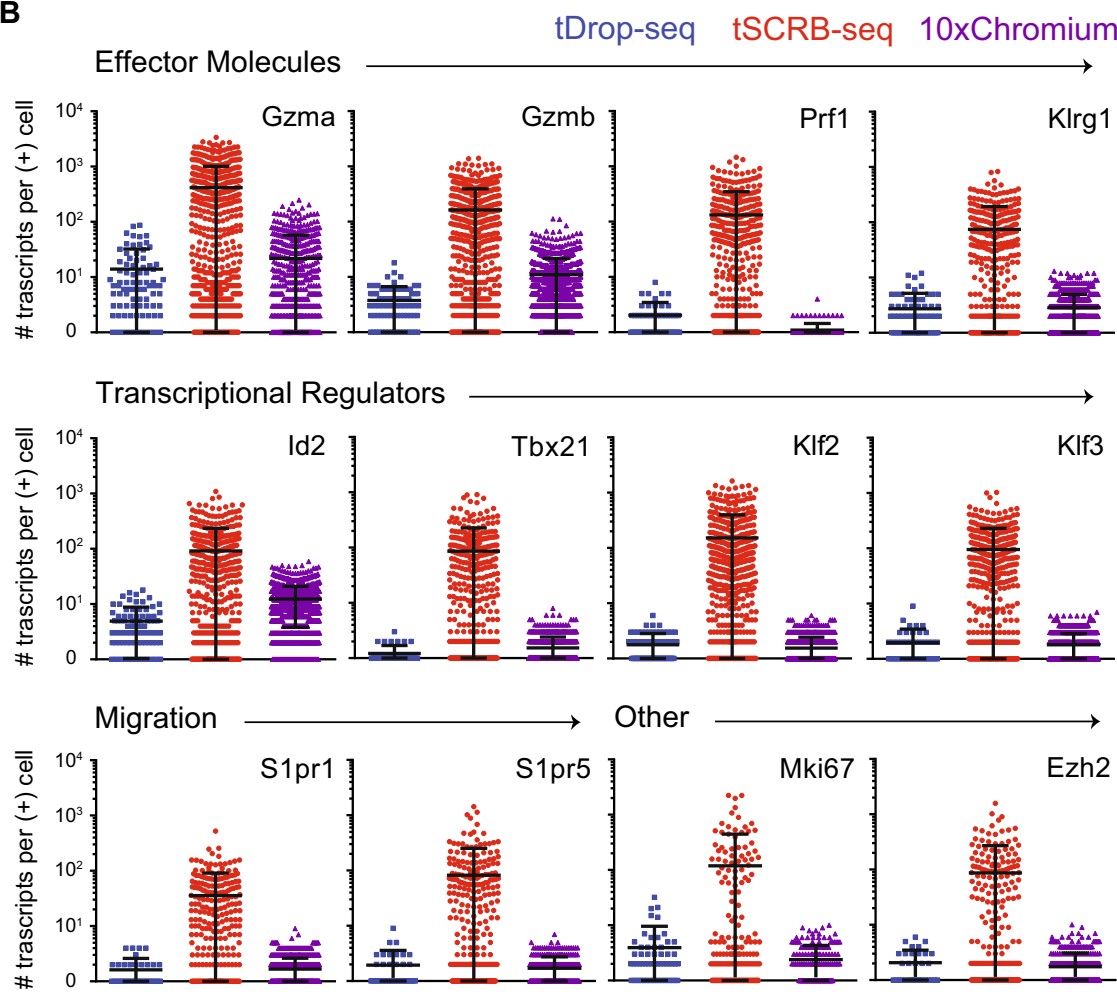

**Fig. 3 The higher transcript yield of tSCRB-seq leads to improved dynamic range of immune gene detection.** Analysis of libraries generated with tDrop-seq and tSCRB-seq from P14 T cells recovered on day 8 post-acute LCMV Armstrong infection, compared to a published 10xChromium dataset with matching experimental set-up[22]. **A** Plot depicting the mean number of detected transcripts (UMIs) per cell among the methods at different sequencing depths (reads mapped to exon regions). **B** Plots depicting the number of captured transcripts of key immune genes per positive cell (cell expressing the respective gene) among the three methods. Each dot represents individual cell. The dot color codes for the method used—blue for tDrop-seq, red for tSCRB-seq, and violet for 10xGenomics. The lines indicate the mean and the standard deviation. Source data are provided as a Source data file.

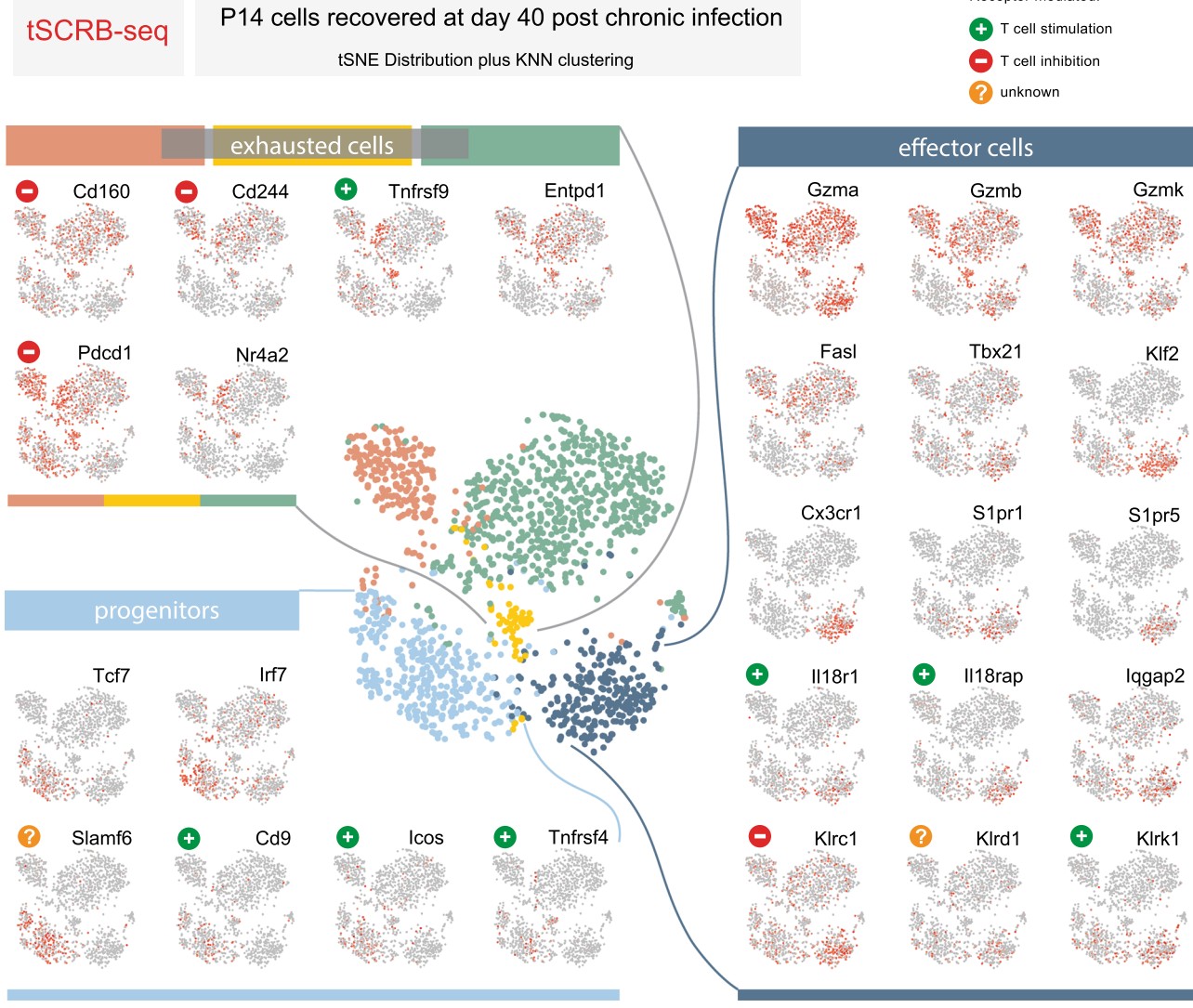

**Fig. 4 tSCRB-seq enables compartment-resolved expression of key co-inhibitory and co-stimulatory receptor targets.** Analysis of published single-cell transcriptomes generated with tSCRB-seq from P14 T cells recovered at day 40 post chronic LCMV c13 WT infection from control or CD4-depleted animals[12]. Each circle represents a single cell. All plots are generated with the use of a non-linear dimensional reduction tSNE (t-distributed Stochastic Neighborhood Embedding). The central plot represents the Seurat-predicted clusters (cell color) depicted over the tSNE. The small side plots represent the expression of key for CD8 T cell differentiation and function immune genes and regulatory receptors depicted over the tSNE.

makes tSCRB-seq perfectly suited for high-depth CTL analysis. This comes at the cost of limited throughput and increased labor intensiveness. Nevertheless, such high-sensitivity approaches like tSCRB-seq have the potential to shed more light on the process of T cell differentiation in health and disease and empower new strategies for targeting challenging immunological diseases. This is demonstrated by the compartment-resolved expression of key co-inhibitory and co-stimulatory receptor targets on CTLs. Finally, we provide a framework for future scRNA-seq protocol optimization for difficult but biologically relevant primary cell types.

## Methods

**Mice**. P14 TCRαβ (CD45.1[+]) transgenic mice were kindly provided by A. Oxenius. Mice were bred and maintained in modified specific pathogen-free facilities of the Technical University of Munich. Experiments were performed with at least 6-week-old mice in compliance with the Technical University of Munich institutional guidelines and were legally approved by the regional veterinary authority "Regierung of Oberbayern." Mice were maintained at a temperature 20–24°C,

humidity 50–70%, and 12-h light cycle with light phase beginning at 5 a.m. and ending at 5 p.m.

**Infections**. In all, $2 \times 10^5$ plaque-forming units (pfu) wild-type LCMV Armstrong or $5 \times 10^6$ pfu wild-type LCMV clone 13 were diluted in phosphate-buffered saline (PBS) and administered into host mice via intraperitoneal or intravenous injection, respectively.

**Purification of mouse and human lymphocytes**. The use of human blood was approved by the ethics committee of Ludwig Maximilian University of Munich. Informed consent was obtained from all participants. Mouse and human lymphocytes were isolated from a mouse spleen and human blood with the use of Lympholyte M (Cedarlane) and Ficoll-Paque PLUS (GE Healthcare) density gradient media, respectively. Cells were washed with media and sorted on BD FACS Fusion (100-micron nozzle, standard operation settings, single-cell purity), where individual cells meeting the gating strategy were sorted in a tube containing media and used immediately for generating Drop-seq single-cell transcriptomes.

**Purification of naive P14 T cells and cell sorting**. Single-cell splenocyte suspensions were obtained by mashing total spleens through a 100-μm nylon cell strainer (BD Falcon) and lysing red blood cells with a hypotonic ACK buffer. Naive

transgenic P14 CD8 T cells were isolated using the mouse CD8+ T cell Enrichment Kit (Miltenyi Biotech, Bergisch-Gladbach, Germany). Surface staining was performed for 40 min at 4 °C in supplemented Dulbecco's modified Eagle's medium (DMEM) media (Gibco, ThermoFisher Scientific) using the following antibodies: anti-CD8a-APC (clone 53-6.7; dilution 1:400; Biolegend), CD4-FITC (clone RM4-4; dilution 1:400; Biolegend), and TCR V alpha 2-PE (clone B20.1; dilution 1:400; eBioscience). Cells were washed twice with media and sorted on BD FACS Fusion (100-micron nozzle, standard operation settings, single-cell purity). Individual cells meeting the gating strategy were sorted in tube containing media (Drop-seq) and used immediately or directly in lysis buffer into individual wells of a low-binding PCR plate (SCRB-seq), which was subsequently spun down, snap-frozen on dry ice, and stored at −80 °C until use. All data were analyzed using FlowJo (TreeStar).

**Purification of activated P14 T cells and cell sorting.** Single-cell splenocyte suspensions were obtained by mashing total spleens through a 100-μm nylon cell strainer (BD Falcon) and red blood cells were lysed with a hypotonic ACK buffer. Naive transgenic P14 CD8 T cells were isolated using the mouse CD8+ T cell Enrichment Kit (Miltenyi Biotech, Bergisch-Gladbach, Germany). In all, $2 \times 10^3$ CD45.1$^+$ P14 TCRα$\beta$ was transferred into naive CD45.2+C57BL/6 mice, followed by their infection. On the day of the sort, single-cell splenocyte suspensions were obtained as described above. Activated transgenic P14 CD8 T cells were isolated using anti-CD45.1 biotin (clone A20; dilution 1:200; eBioscience)/anti-biotin-conjugated microbeads and MACS cell separation (Miltenyi Biotech, Bergisch-Gladbach, Germany). Surface staining was performed for 40 min at 4 °C in supplemented DMEM media (Gibco, ThermoFisher Scientific) using the following antibodies: anti-CD8a-PerCP-Cy5.5 (clone 53-6.7; dilution 1:400; Biolegend), CD4-FITC (clone RM4-4; dilution 1:400; Biolegend), CD45.1-APC (clone A20; dilution 1:400; eBioscience), and CD45.2-eFluor 450 (clone 104; dilution 1:400; eBioscience). Cells were washed twice with media and sorted on BD FACS Fusion (100-micron nozzle, standard operation settings, single-cell purity). Individual cells meeting the gating strategy were sorted in tube containing media (Drop-seq) and used immediately or directly in lysis buffer into individual wells of a low-binding PCR plate (SCRB-seq), which was subsequently spun down, snap-frozen on dry ice, and stored at −80 °C until use. All data were analyzed using FlowJo (TreeStar).

**Generation of single-cell transcriptomes with Drop-seq.** The original Drop-seq protocol was performed based on Macosko and colleagues[16]. The hardware used included an inverted microscope, three syringe pumps (Legato 100, KD Scientific), a magnetic stirrer (710D2, VP Scientific), a magnetic stirring disc to keep the barcoded beads suspended (772DP-N42-5-2, VP Scientific), a Peqlab PerfectBlot hybridization oven, and a thermal cycler. The barcoded Drop-seq oligo-dT beads (MACOSKO-2011-10; ChemGenes Corporation) were washed once with 30 ml pure ethanol, twice with 30 ml TE-TW buffer (10 mM 1 M Tris pH 8.0; 1 mM EDTA, and 1% Tween-20), resuspended in 20 ml TE-TW buffer, passed through 100-μm nylon cell strainer (BD Falcon), counted with a Fuchs-Rosenthal hemocytometer, and stored at 4 °C for up to 6 months. The needed quantity of beads was aliquoted and resuspended in 2× lysis buffer (6% Ficoll MP-400; 0.2% Sarkosyl; 20mM EDTA; 200 mM Tris pH 7.5, and 50 mM dithiothreitol) at a final concentration 120,000 beads/ml. The sorted cells were diluted with PBS supplemented with 1% bovine serum albumin at final concentration of 100,000 cells/ml. The 20 ml syringe containing droplet generation oil (186-4006, Bio-Rad) was mounted on the oil syringe pump and connected to the Drop-seq PDMS device (Nanoshift). The cells were loaded in 3 ml syringe (309657, BD), mounted on the cell syringe pump, and connected to the Drop-seq PDMS device (Nanoshift). The beads resuspended in 2× lysis buffer were loaded in 3 ml syringe (309657, BD) together with a magnetic stirring disk, mounted on the bead/lysis syringe pump in proximity to the magnetic stirrer, and connected to the Drop-seq PDMS device (Nanoshift). The droplet generation was performed at 15,000 μl/h oil flow, 3000 μl/h cell flow, and 3000 μl/h bead flow. After removal of the oil, 30 ml 6× SSC (15557-044, Thermo Fisher Scientific) and 1 ml perfluorooctanol (647-42-7, Sigma-Aldrich) were added, and the generated droplets were broken by manual vertical shaking. The beads were washed once with 30 ml 6× SSC, twice with 1 ml 6× SSC, and once with 1 ml 5× reverse transcription buffer (EP0753, Thermo Fisher Scientific). The beads were resuspended in 200 μl of reverse transcription master mix (1× Maxima H− RT Buffer—EP0753, Thermo Fisher Scientific; 4% Ficoll MP-400; 1 mM Advantage UltraPure PCR Deoxynucleotide Mix—639125, Clontech; 1 U/μl NxGen RNAse Inhibitor—30281-2, Lucigen; 2,5 μM Drop-seq Template Switching Oligo—Eurogentec; 10 U/μl Maxima H Minus RT—EP0753, Thermo Fisher Scientific), incubated in the hybridization oven for 30 min at room temperature with rotation, and then for 90 min at 42 °C with rotation. All Drop-seq primer sequences used in this study are available in (Supplementary Table 3). The beads were washed once with 1 ml TE-SDS (10 mM Tris pH 8.0; 1 mM EDTA; 0.5% SDS), twice with 1 ml TE-TW buffer, and once with 1 ml 10 mM Tris pH 8. The beads were resuspended in 200 μl exonuclease reaction mix (1× Exonuclease I Reaction Buffer—B0293S, New England Biolabs; 1 U/μl Exonuclease I—M0293S, New England Biolabs) and incubated in the hybridization oven for 45 min at 37 °C with rotation. The beads were washed once with 1 ml TE-SDS, twice with 1 ml TE-TW buffer, once with 1 ml molecular-grade water, and resuspended in 1 ml molecular-grade water. The beads were counted with a Fuchs-Rosenthal hemocytometer and 2000 beads were apportioned per PCR reaction. The apportioned

beads were resuspended in 50 μl PCR master mix (1× Kapa HiFi Hotstart Readymix—KK2602, Kapa Biosystems; 0.8 μM Drop-seq SMART PCR Primer) and incubated in a thermal cycler using the following program—heated lid at 100 °C, 3 min 95 °C, 4 cycles (20 s 98 °C, 45 s 65 °C, 3 min 72 °C), 14 cycles (20 s 98 °C, 20 s 67 °C, 3 min 72 °C), 7 min 72 °C, and hold at 4 °C. The barcoded single-cell amplicons were purified with the use of (0.6×) AMPure XP beads (A63881, Beckman Coulter). The quality and quantity of the resulting amplicon was assessed with the use of Agilent High Sensitivity DNA Kit (5067-4626, Agilent). One nanogram of the resulting amplified cDNA was used for library preparation with the Illumina Nextera XT DNA Library reagents (FC-131-1024, Illumina). The Nextera XT N5 index primer was substituted with a Drop-seq custom N5 primer, which was used along with Nextera XT N7 index primer. After PCR amplification of the fragmented libraries, the samples were purified with (0.6×) AMPure XP beads and eluted in 10 μl of molecular-grade water. The quality of the resulting library was assessed with the use of Agilent High Sensitivity DNA Kit (5067-4626, Agilent). The library quantification was performed based on the Illumina recommendations (SY-930-1010, Illumina) with the use of KAPA SYBR FAST qPCR Master Mix (KK4600, Kapa Biosystems). The libraries were sequenced on Illumina HiSeq 2500 system at the following conditions—rapid run, paired-end, 20 bp read 1, 45 bp read 2, single-indexed sequencing resulting in 0.5 million reads per single cell. Due to the use of Drop-seq custom N5 primer, the Illumina HP10 read 1 primer was replaced with Drop-seq Custom Read 1 Primer following the manufacturer's recommendations. In the process of Drop-seq protocol optimization described in this article, the following substitutions were tested. Replacement of the 0.2% Sarkosyl with 0.2% Igepal (I8896-50ML, Sigma-Aldrich; final concentration in the droplet 0.1%). Addition of 0.5 M NaCl (S3014, Sigma-Aldrich) to the lysis buffer. Replacement of the 3' most rG in the TSO with a locked nucleic acid base (3'LNA). Replacement of the 10 U/μl Maxima H Minus RT for 10 U/μl SuperScript IV RT (18090010, ThermoFisher) or 10 U/μl SMARTScribe RT (639536, Takara Bio). Addition of 4% Ficoll PM 400 (GE17-0300-10, GE Healthcare) to the PCR master mix as a macromolecular crowding.

**Generation of single-cell transcriptomes with SCRB-seq.** The original SCRB-seq protocol was performed based on Soumillon and colleagues[17] with some modifications necessary for working with primary T cells. The single-cell RNA was purified with (2.2×) RNAClean XP beads (A63987, Beckman Coulter) before reverse transcription. The RNA was eluted in 2.8 μl of molecular-grade water supplemented with 1.42 U/μl NxGen RNAse Inhibitor (30281-2, Lucigen). In all, 1.2 μl of unique tSCRB Barcoded Oligo-dT Primer was added to each well of the PCR plate at a final concentration of 2.4 μM during the reverse transcription. The single-cell plates were incubated on a thermal cycler using the following program— heated lid at 105 °C, 3 min at 72 °C, hold at 4 °C. The single-cell transcriptomes from the optimization experiments were barcoded with tSCRB Barcoded Oligo-dT Primer Plate v1 (Supplementary Table 4). The single-cell transcriptomes from P14 T cells recovered at day 8 post LCMV Armstrong infection were barcoded with tSCRB Barcoded Oligo-dT Primer Plate v2 (Supplementary Table 5). Currently, we recommend the use of tSCRB Barcoded Oligo-dT Primer Plate v3 (Supplementary Table 6), which has the most optimized cellular barcode distance. The SCBR-seq primers used in this study are available in Supplementary Table 3. Three microliters of reverse transcription master mix (1× Maxima H Minus Buffer—EP0753, Thermo Fisher Scientific; 10 U/μl Maxima H Minus Reverse Transcriptase— EP0753, Thermo Fisher Scientific; 1 mM Advantage dNTPs Mix—639125, Takara; 1.42 U/μl NxGen RNAse Inhibitor—30281-2, Lucigen; and 1.2 μM SCRB-seq TSO) were added to each well, and the plates were incubated on a thermal cycler using the following program—heated lid at 105 °C, 90 min at 42 °C, 15 min at 72 °C, hold at 4 °C. Eighteen microliters of reverse transcription master mix (1× KAPA HiFi HotStart ReadyMix—7958935001, KAPA Biosystems; 0.48 μM SCRB-seq SMART PCR Primer) were added to each well, and the plates were incubated on a thermal cycler using the following program—heated lid at 100 °C, 3 min at 98 °C, 20 cycles (20 s at 98 °C, 30 s at 65 °C, 6 min at 72 °C), 5 min at 72 °C, hold at 4 °C. The single-cell amplicons from each plate were pooled together and double purified with the use of (0.6×) AMPure XP beads (A63881, Beckman Coulter). The quality and quantity of the resulting pooled plate amplicon was assessed with the use of Agilent High Sensitivity DNA Kit (5067-4626, Agilent). One nanogram of the resulting amplified cDNA was used for library preparation with the Illumina Nextera XT DNA Library reagents (FC-131-1024, Illumina). The Nextera XT N5 index primer was substituted with a SCRB-seq custom N5 primer, which was used along with Nextera XT N7 index primer. After PCR amplification of the fragmented libraries, the samples were double purified with (0.6×) AMPure XP beads and eluted in 10 μl of molecular-grade water. The quality of the resulting library was assessed with the use of Agilent High Sensitivity DNA Kit (5067-4626, Agilent). The library quantification was performed based on the Illumina recommendations (SY-930-1010, Illumina) with the use of KAPA SYBR FAST qPCR Master Mix (KK4600, Kapa Biosystems). The libraries were sequenced on Illumina HiSeq 2500 system at the following conditions—rapid run, paired-end, 16 bp read 1, 49 bp read 2, single-indexed sequencing resulting in 0.5 (naive and day 8 acute infection datasets) or 1.0 (day 40 chronic infection dataset) million reads per single-cell. Due to the use of SCRB-seq custom N5 primer, the Illumina HP10 read 1 primer was replaced with SCRB-seq Custom Read 1 Primer following the manufacturer's recommendations. In the process of SCRB-seq protocol optimization described in this article, the

following substitutions were tested. Replacement of the 1:500 dilution of Phusion HF buffer supplemented with Proteinase K for cell lysis with TCL buffer (1031576, Qiagen) supplemented with 1% β-mercaptoethanol (M3148-25ML, Sigma-Aldrich) or 0.2% Triton X-100 detergent (T9284-100ML, Sigma-Aldrich). Replacement of the 3' most rG in the TSO with a locked nucleic acid base (3'LNA). A step-by-step final version of the T cell-tailored single-cell RNA barcoding and sequencing (tSCRB-seq) protocol is available via Protocol Exchange [https://doi.org/10.21203/rs.3.pex-1289/v1].

**Single-cell RNA-seq data pre-processing and analysis**. DropSeqPipe v0.4 was used for raw data processing[28]. Parameters are provided in the configuration file on the repository GSE163089. Cutadapt v1.16 was used for trimming[29]. Trimming and filtering was done on both fastq files separately. Reads with a missing pair were discarded using bbmap v38.22. STAR v2.5.3a[30] was used for mapping to annotation release #91 and genome build #38 from *Mus musculus* (Ensembl). Multi-mapped reads were discarded. Dropseq_tools v1.13 was used for demultiplexing and file manipulation[16]. A whitelist of cells barcodes with minimum distance of 3 bases was used. Cell barcodes and UMIs with a hamming distance of 1 and 2, respectively, were corrected. For the cell clustering, 2000 genes were selected for the downstream analysis using depth-adjusted negative binomial model by M3Drop[31]. The Seurat package was used for further processing[16]. Cell subpopulations were detected by Louvain clustering with top 5 principle components and K.para = 150.

**Statistics and reproducibility**. The Drop-seq human and mouse lymphocyte mixing experiment was repeated three times. The baseline Drop-seq protocol and each separate Drop-seq optimization with naive P14s were repeated once. The final tDrop-seq protocol with P14 T cell recovered at day 8 post LCMV Arm infection was repeated two times. The baseline SCBR-seq protocol and each separate SCBR-seq optimization were repeated once. The final tSCRB-seq protocol with P14 T cell recovered at day 8 post LCMV Arm infection was performed with ten plates processed on three different days, each plate representing an individual replicate.

**Reporting summary**. Further information on research design is available in the Nature Research Reporting Summary linked to this article.

## Data availability

Datasets generated during the current study are available in the Gene Expression Omnibus (GEO) repository with accession code GSE163089. The previously published single-cell datasets analyzed in the current study are available in the Gene Expression Omnibus (GEO) with the following accession codes: 10xGenomics single-cell datasets of naive P14 T cells and P14 T cells recovered on day 8 post LCMV Armstrong[22]—GSE131535; 10xGenomics single-cell dataset of P14 T cells recovered on day 30 post LCMV clone 13 infection[13]—GSE129139. Any other relevant data are available from the authors upon reasonable request. Source data are provided with this paper.

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

## Acknowledgements

We thank Wolfgang Enard and Christoph Ziegenhain for the initial guidance in selecting between the existing range of single-cell RNA sequencing protocols and Tobias Herbinger, Brigitte Dötterböck, and Waltraud Schmid for technical assistance and animal husbandry. The work was supported by the European Research Council starting and consolidator grants (ProtecTC and ToCCaTa) and from the German Research Foundation (SFB1054).

## Author contributions

K.K. and D.Z. designed experiments. K.K. performed experimental work, including protocol optimizations and library preparations, and collected and analyzed results. P.R. developed the single-cell analysis pipeline and performed bioinformatical analysis. M.W. performed bioinformatical analysis. C.W. performed experimental work. M.W.P. and M.D. provided expertise in protocol optimization and bioinformatics, respectively. K.K. prepared the figures. D.Z. and K.K. wrote the manuscript. D.Z. conceived the study and supervised the project.

## Funding

## Competing interests

P.R. currently works for 10xGenomics. The single-cell analysis pipeline developed by him and the analysis performed were part of his graduate studies, before joining the team of 10xGenomics. The other authors declare no competing interests.
