## [Peer Review File · Nature Communications]

Reviewers' Comments:

Reviewer #1:

Remarks to the Author:

This manuscript by Kanev et al describes technical improvements in methods used to measure gene expression at the single cell resolution. In particular, the study focuses on enhancements in both droplet-based and plate-based techniques to study T cells, either naive or in response to viral infections. The authors suggest that such amelioration could be of potential interest to increase the power of computational analyses aimed at deciphering the heterogeneity and transcriptional network involved in cytotoxic CD8 T cells.

The study shows that modifications of current protocols can improve the detection of genes for notoriously difficult cell types like T lymphocytes. By changing conditions of cell lysis and RNA isolation, the authors show a modest improvement in the gene capture for drop-based methods while providing evidences of an important increase in transcript detection for the plate-based method. In this regard, although the biological insights are here limited, the study by Kanev et al will be valuable for the single-cell and immunology community.

However, the current study has a number of important experimental and analytical limitations that will need to be addressed in order to strengthen the validity of the conclusions.

1/To compare the metrics of each methods (current, improved or published) the authors need to take into account the depth of sequencing for individual cells. Although it may not be critical for all experiments performed in house that apparently have identical sequencing depth (see point 3), it become critical when compared to published datasets (Chen et al for naive and acute). Comparing "performance" metrics and downstream analyses can only be valid when the numbers of reads per cells are similar. The authors should therefore downsample their own datasets to match the sequencing depth reached by previously published datasets.

Comparing tDROPseq metrics to the naive dataset in Chen et al will give a side-by-side comparison. As these naive populations are expected to be quite homogeneous it would be hard to use both methods to further explore their performances in term of clustering and/or network analyses. However, it will be informative to have the relative transcript levels for genes expressed at steady state in this population (Cd8a, Cd8b, Tcf7, Sell, Runx3, Ccr7 ...).

Similarly, tSCRB-seq dataset will need to be analyzed side-by-side with published datasets after sequencing depth normalization. Further analyses and comparison with published datasets from similar experimental settings (both acute from Chen et al and chronic from PMID 30778252 for example) will need to be performed to demonstrate the tSCRB-seq allows better computational analyses including detection of clusters and gene signatures (see point 4).

2/Concerning the results of the tSCRB-seq method, although the numbers of transcripts are massively increased compared to the 10X platform, it is unclear as to why this does not result in substantially higher gene numbers. This would suggest that the improvement only increases the number of transcripts for genes already detected. Although this will be important to have a more dynamic range in gene expression for downstream analysis, the author should emphasize whether this increase in UMI/gene distributes evenly across all genes. In fact, this improvement will only ameliorate analyses if these new UMIs aligned to functionally relevant genes rather than housekeeping or structural genes. This may be what the authors suggest with the "md top50 genes" metric in Table 1 but this should be clarified and expanded upon. Having normalized numbers of transcripts for known markers and transcription factors (both highly expressed and less abundant) detected across these methods will be an important validation that the method could indeed increase the analytical power of downstream computational analyses.

3/Figure 3 shows the transcriptional landscape of CD8 T cells in response to chronic infection, either in the absence or presence of CD4 T cells. This dataset appears very similar that one previously published (Kanev et al., PNAS 2019). The authors should make clear if this dataset is already published especially because there seems to be discrepancies in the described methods.

When Kanev et al., PNAS 2019 mentioned that "[...]single-indexed sequencing resulting in 1 million reads per single-cell" in supplemental methods, the current manuscript describes "single-indexed sequencing resulting in 0.5 million, line 355". The authors should clarify this point and perform analysis after downsampling to a common sequencing depth.

4/Regardless of the issue of sequencing depth, it is not clear which new biological insights the tSCRB-seq methods brings in the current manuscript. Is there any new markers or gene expression signatures that would not have been captured by the 10X methods? The dataset presented in Figure 3 and the one with the acute settings using the tSCRB-seq need to be compared to published datasets with a relatively similar experimental system.

Minor points

The number of clonally expanded daughter cells has been shown to be on average in the 1000-10000 range by multiples studies on both CD8 (PMID 23493420 and 23493421) and CD4 (PMID 26823430 and 28746867). The statement line 42-44 should be referenced and put in perspective with these other studies.

Important references regarding the role of Tcf1 in chronic settings and tumors are missing (PMID: 28018990 , 27501248 among others)

Figure 1E calling incorrect line 138 and 472

Reviewer #2:

Remarks to the Author:

The authors addressed a technical issue in the field of single-cell sequencing concerning the low capture efficiency of mRNAs, which is even more relevant for immune cells that have, on average, lower transcript abundance compared to other cell types. They performed a technical setup and improved Drop-seq and SCBR-seq to make them more suitable and sensitive for the studies on T cells and in particular on CTLs.

Despite the experimental part is technically sound, I feel that the manuscript in the present form does not provide enough biological insights.

Dear Editor and Reviewers,

We thank you very much for your comprehensive feedback and helpful suggestions. In response to the points raised, we performed additional analyses which are now included in the revised version of our manuscript. These include:

- Downsampling of the number of reads per cell to demonstrate the transcript saturation point and the superior performance of tSCBR-seq compared to droplet-based methods.
- Demonstrating that the gain in transcripts obtained with tSCBR-seq increases the dynamic range at which immune genes are detected.
- Comparing the cluster resolution obtained with tSCBR-seq in advanced chronic infection to a comparable published dataset generated with the 10xChromium platform.

All significant changes and new information are underlined in the revised manuscript text.

Please note that some of the figure and table numbers have been changed during the revision to allow the accommodation of new analyses and to improve the clarity of the represented data.

We hope that the reviewers agree that the experiments and changes borne out of their comments have substantially strengthened our manuscript and clarified the points that were raised.

Referee #1 (Remarks to the Author):

We would like to thank the reviewer for her/his supportive comment, for stating that “the study by Kanev et al will be valuable for the single-cell and immunology community”, and for the provision of very helpful ideas to revise our manuscript.

1) “To compare the metrics of each methods (current, improved or published) the authors need to take into account the depth of sequencing for individual cells. Although it may not be critical for all experiments performed in house that apparently have identical sequencing depth (see point 3), it become critical when compared to published datasets (Chen et al for naive and acute). Comparing “performance” metrics and downstream analyses can only be valid when the numbers of reads per cells are similar. The authors should therefore downsample their own datasets to match the sequencing depth reached by previously published datasets.”

>>> We completely agree that the sequencing depth can have a drastic impact on the performance metrics. All 10xChromium datasets that we included in our analysis were sequenced in line with the

manufacturer’s recommended sequencing depth of 50 000 reads per cell for PBMCs (which include T cells). The manufacturer showed that saturation of the number of captured genes and transcripts is reached at this sequencing depth (10xGenomics CG000148 Rev A Technical Note – Resolving Cell Types as a Function of Read Depth and Cell Number; the figures included are obtained from

Figure 3B from the manuscript

the datasheet). To exclude any doubt, we have performed down-sampling by our own (**Figure 3B**). We demonstrate that both 10xChromium and tDrop-seq saturate at 50 000 reads per cell, thus as previously shown by 10xGenomics a deeper sequencing is not necessary. In contrast, tSCRB-seq does not reach saturation at this sequencing depth because of the more than 10-fold increased transcript capturing efficacy. Thus, we recommend sequencing the tSCRB-seq libraries to at least 200 000 reads per cell. Nevertheless, if similar coverage of each

transcript (mapped reads per UMI) should be achieved, the tSCRB-seq library should be sequenced at depth equal to the fold increase in capturing compared to 10xChromium (12 x 50 000 reads per cell). Moreover, we have observed that tSCRB-seq captures significantly more transcripts per cell even when downsampled to the sequencing depth of 10xChromium (~17 000 vs ~6500 respectively; **Figure 3B**). As suggested by the reviewer, we compared the performance of the three methods with (**Supplementary Table 2**) and without read down sampling (**Figure 3**) and included the data into the revised manuscript.

2) “Concerning the results of the tSCRB-seq method, although the numbers of transcripts are massively increased compared to the 10X platform, it is unclear as to why this does not result in substantially higher gene numbers. This would suggest that the improvement only increases the number of transcripts for genes already detected. Although this will be important to have a more dynamic range in gene expression for downstream analysis, the author should emphasize whether

this increase in UMI/gene distributes evenly across all genes. In fact, this improvement will only ameliorate analyses if these new UMIs aligned to functionally relevant genes rather than housekeeping or structural genes. This may be what the authors suggest with the "md top50 genes" metric in Table 1 but this should be clarified and expanded upon. Having normalized numbers of transcripts for known markers and transcription factors (both highly expressed and less abundant) detected across these methods will be an important validation that the method could indeed increase the analytical power of downstream computational analyses."

>>> We agree with the reviewer that this is an interesting point that puzzled us for a while as well. Nonetheless, we have revised our data presentation to better highlight that the gain in transcript numbers obtained with tSCBR-seq significantly enhances the dynamic range of immune gene detection, which are otherwise found at very low quantities in single cells. We demonstrate this with and without read down sampling in the revised **Supplementary Table 2** and **Figure 3C**.

Figure 3C from the manuscript

3) "Figure 3 shows the transcriptional landscape of CD8 T cells in response to chronic infection, either in the absence of presence of CD4 T cells. This dataset appears very similar that one previously published (Kanev et al., PNAS 2019). The authors should make clear if this dataset is already published especially because there seems to be discrepancies in the described methods. When Kanev et al., PNAS 2019 mentioned that "[...]single-indexed sequencing resulting in 1 million reads per single-cell" in supplemental methods, the current manuscript describes "single-indexed sequencing resulting in 0.5 million, line 355". The authors should clarify this point and perform analysis after downsampling to a common sequencing depth."

>>> We would like to apologize for leaving this point unclear. This is indeed a previously published dataset and we are highlighting this now more clearly in the main manuscript text and in the figure legend. In contrast to the day 8 Armstrong tSCBR-seq dataset which was only generated for protocol comparison, the experimental day 40 clone 13 tSCBR-seq dataset was sequenced to a depth of 1 million reads per cell. These differences are now clearly stated in the materials and methods section of the manuscript.

4) “Regardless of the issue of sequencing depth, it is not clear which new biological insights the tSCRB-seq methods brings in the current manuscript. Is there any new markers or gene expression signatures that would not have been captured by the 10X methods? The dataset presented in Figure 3 and the one with the acute settings using the tSCRB-seq need to be compared to published datasets with a relatively similar experimental system.”

>>> We have compared the day 40 chronic tSCRB-seq dataset to the day 30 chronic 10xChromium dataset previously published by Zander and colleagues (**Supplementary Figure 4**). Though this is a slightly complex task as the experimental setting is not completely identical, we would like to highlight that tSCRB-seq provides more detailed and graded differential expression of key immune genes among the predicted by us clusters. This particularly affected key transcriptional factors (e.g. Tbx21 and Irf7) and inhibitory receptors (Pdc1, Cd160 and Entpd1).

Minor points:

1) “The number of clonally expanded daughter cells has been shown to be on average in the 1000-10000 range by multiples studies on both CD8 (PMID 23493420 and 23493421) and CD4 (PMID 26823430 and 28746867). The statement line 42-44 should be referenced and put in perspective with these other studies.”

>>> We have corrected the inaccuracy and added the suggested references.

2) “Important references regarding the role of Tcf1 in chronic settings and tumors are missing (PMID: 28018990 , 27501248 among others)”

>>> We have added the references.

3) “Figure 1E calling incorrect line 138 and 472”

>>> We have corrected the mistake.

Referee #2 (Remarks to the Author):

We would like to thank the reviewer for highlighting that the “experimental part [of the manuscript] is technically sound”.

1) “I feel that the manuscript in the present form does not provide enough biological insights”

>>> We completely understand the point. We would like to kindly underline that our intention with this manuscript is to draw attention to the CTL-tailored tSCRB-seq as a powerful solution for single-

gene expression measurements with high dynamic range. We have successfully utilized tSCBR-seq to shed length on several key immunological question, one which is to deconvolute the heterogeneity of effector populations in chronic infection (PMID: 31530725), thus we are convinced in the protocol performance. Unfortunately, the size and structure of this paper and manuscripts do not allow in depth showcase and discussion of the advantages of tSCBR-seq. Moreover, we see this manuscript as a model for single-cell RNA sequencing protocol tailoring for difficult, but biologically relevant cell types like primary CD8 T cells. Thus, we believe that this article will be highly cited and influential even beyond the field of immunology.

Reviewers' Comments:

Reviewer #1:

Remarks to the Author:

The authors addressed most of the points raised by the reviewers. Although the biological findings of this study are rather limited, the technological improvements provided by the tSCBR-seq method would be of great interest for the immunology community and beyond.